# Electrochemical Behavior and Surface Conductivity of C/TiC Nanocomposite Coating on Titanium for PEMFC Bipolar Plate

**Wei Meng [1,2], Haifeng Zhu [1,2], Xiaopeng Wang [3], Guanghui Li [4], Yingze Fan [4], Deen Sun [5] and Fantao Kong [1,2,*]**

[1] State Key Laboratory of Advanced Welding and Joining, Harbin Institute of Technology, Harbin 150001, China; 13840045647@163.com (W.M.); 18846161834@163.com (H.Z.)

[2] School of Materials Science and Engineering, Harbin Institute of Technology, Harbin 150001, China

[3] Center of Analysis and Measurement, Harbin Institute of Technology, Harbin 150001, China; wangxiaopeng@hit.edu.cn

[4] School of Materials Science and Engineering, Chongqing University, Chongqing 400044, China; LGH_CQU@163.com (G.L.); 202009131193@cqu.edu.cn (Y.F.)

[5] School of Materials and Energy, Southwest University, Chongqing 400715, China; sundeen0909@swu.edu.cn

[*] Correspondence: kft@hit.edu.cn

**Abstract:** In this study, a C/TiC nanocomposite coating has been prepared by magnetron sputtering technology and vacuum heat treatment technology on a titanium surface, which is used for bipolar plates (BPs) in a proton exchange membrane fuel cell (PEMFC). This prepared C/TiC nanocomposite coating was characterized by X-ray diffraction (XRD), scanning electron microscopy (SEM), X-ray photoelectron spectroscopy (XPS), Raman spectroscopy, electrochemical testing and interfacial contact resistance (ICR). The results show that a C/TiC nanocomposite coating consists of a single C surface layer (~28.88 nm) and TiC interface layer (~19.5 nm). In addition, compared with commercially pure titanium substrate ($i_{corr}$ = 345.10 μA cm$^{-2}$), the corrosion resistance of a C/TiC nanocomposite coating ($i_{corr}$ = 0.74 μA cm$^{-2}$) was greatly improved in 0.5 M $H_2SO_4$ + 5 ppm HF solution at 80 °C. The corrosion current density ($i_{corr}$) decreased 3 orders of magnitude in a simulated cathodic environment. Moreover, the interfacial contact resistance of a C/TiC nanocomposite coating is 2.34 mΩ cm$^2$ under 1.4 MPa compaction force, which is much lower than that of raw CP Ti (38.66 mΩ cm$^2$).

**Keywords:** PEMFC; titanium bipolar plate; nanocomposite coating; corrosion resistance; interfacial contact resistance

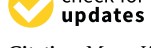



## 1. Introduction

In recent years, the fuel cell has experienced rapid development in various fields, because of its advantages of being pollution-free and highly energy efficient [1–3]. Among them, proton exchange membrane fuel cells (PEMFCs) are considered to be the next generation of energy equipment, due to their advantages of high energy conversion efficiency, zero-emissions, fast start-up, as well as low operating noise [2,4,5]. PEMFCs have a sandwich-like structure, which consists of shells, bipolar plates, and membrane electrode components (gas diffusion layer-GDL, proton exchange membrane-PEM and catalyst layer-CL) [6,7], as shown in Figure 1. When PEMFCs are working, hydrogen and oxygen gases are used as fuels, which go into the anode and cathode of the cell, respectively [1,7,8]. Under the action of the catalyst, the $H_2$ change to $H^+$ ions at the anode and then pass through the PEM to react with $O_2$ at the cathode. The water ($H_2O$) and energy (electricity and heat) are released after that [2]. In accordance with to several descriptions in the literature [1,2,7,8], the schematic diagram of the working principle is shown in Figure 1.

$$\text{Anode Reaction}: 2H_2 \rightarrow 4H^+ + 4e^- \tag{1}$$

$$\text{Cathode Reaction}: O_2 + 4H^+ + 4e^- \rightarrow 2H_2O \tag{2}$$

$$\text{Summary Reaction}: 2H_2 + O_2 \rightarrow 2H_2O \qquad (3)$$

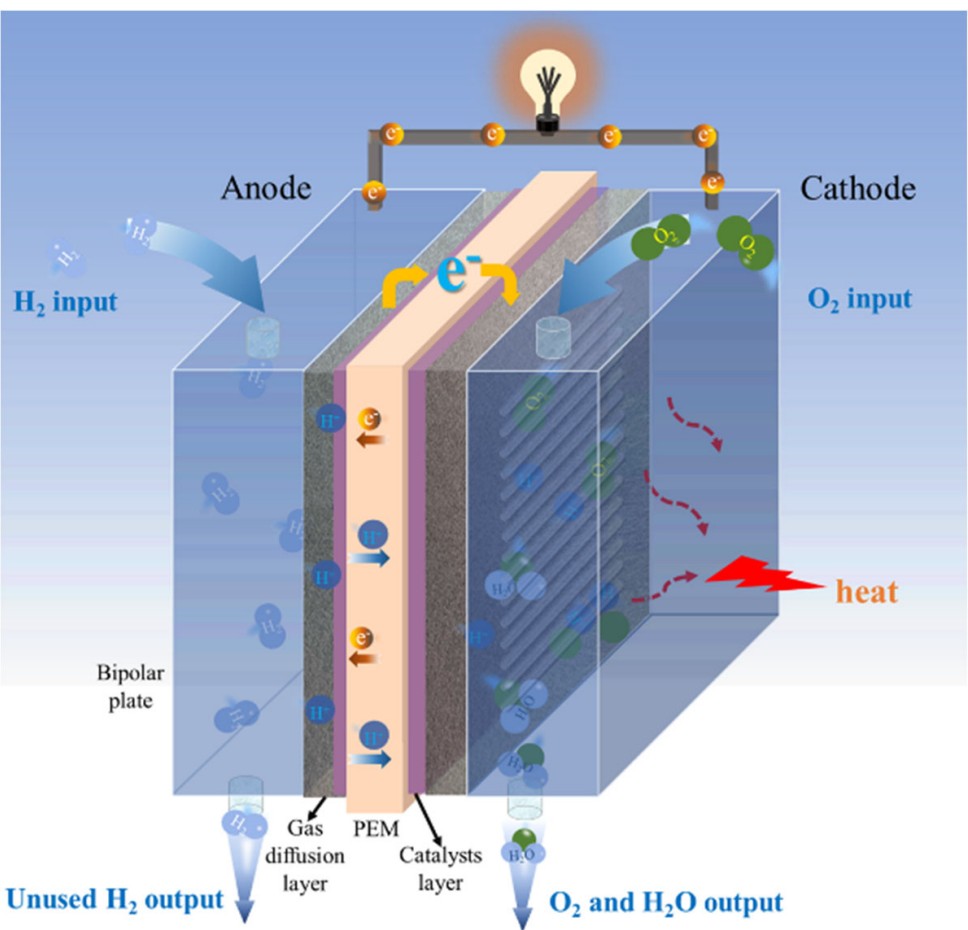

**Figure 1.** The schematic diagram operation of PEMFCs.

In PEMFCs, bipolar plates contributed 60~80% of the weight of the PEMFC stack, 30~40% of the cost and almost occupied the entire volume of the entire battery [9,10], which is considered to be a key component of PEMFCs. Therefore, the ideal bipolar plate must achieve the characteristics of good air tightness, wettability, mechanical properties, electrical conductivity, corrosion resistance and low cost.

Traditional graphite bipolar plates were usually used as battery plate materials because of their good electrical conductivity and corrosion resistance. However, the many disadvantages of low brittleness, poor air tightness and high manufacturing process costs limited their mass production [11,12]. Compared with graphite bipolar plates, the properties of conductivity and mechanical strength are outstanding for stainless steel bipolar plates [13,14], but their high weight density does not meet the requirements of energy efficiency. In addition, stainless steel has terrible corrosion resistance in an acidic solution; the metal ions $Fe^{2+}$ $Cr^{2+}$ and $Ni^{2+}$ are released from substrate to solution during the working process, which can reduce the membrane performance and poison the catalysts [15]. Although corrosion resistance can be improved by surface modification or adding a coating to stainless steels surfaces [7,14,16–20] using different technology, i.e., PVD [4], CVD [21], arc ion plating technology [19], magnetron sputtering [22–24], et al., the complex coating process increases its cost in mass production.

Compared with stainless steel, titanium and titanium [1,25,26] alloy have the obvious advantages of light weight, great corrosion resistance and electrical conductivity. Unfortunately, while the passive film formed on the surface can protect the substrate from corrosion in the PEMFC environment, in the meantime, it will enormously increase the ICR between

the surface and gas diffusion layer (GDL) of bipolar plates and further reduce the battery output power [27]. To address this drawback, various types of coating materials, such as noble metals (Au [28], Ag [29,30], Pt), non-noble metals (Zr [31]), metal nitrides (TiN [32], CrN [33], ZrCN [34]), metal carbonitride coating (TiC [8], TiCrC [35], NbCrC [36]) and amorphous carbon ($\alpha$-C [37]) were prepared by different surface modification technologies. Up to now, there have been no relevant reports on the preparation of nanocomposite coatings through the combination of magnetron sputtering technology and vacuum heat treatment technology. In this work, in order to reduce costs and increase performance for BPs, a C/TiC nanocomposite coated titanium bipolar plate was prepared using a combination of magnetron sputtering technology and vacuum heat treatment technology. It has such characteristics with a single C surface layer using magnetron sputtering technology and a TiC interface layer using magnetron sputtering technology and vacuum heat treatment technology. This prepared C/TiC nanocomposite coating titanium bipolar plate exhibited good electrical conductivity and corrosion resistance in a simulated PEMFC cathodic environment.

## 2. Experiments Details

### 2.1. Preparation of the Nanocomposite Coating

Commercial pure titanium TA1 (CP Ti) was used as the bipolar plate substrate. The CP Ti substrate was cut into pieces with size of 30 mm × 30 mm × 0.1 mm, and was immersed in a 5 vol% HF solution for 30 s in order to remove the oxidized layer. Then the CP Ti substrate was cleaned ultrasonically with deionized water and alcohol for 15 min to remove organic substances, and dried at room temperature. Next, the Ti substrate was put into the vacuum chamber of the deposition system. A DC power source was applied to a graphite target (purity 99.99%) to deposit a single C layer on the CP Ti substrate by magnetron sputtering technology. Prior to the deposition, the vacuum chamber of the deposition system was vacuumized until the pressure was lower than $1 \times 10^{-3}$ Pa, then the pressure was maintained at approximately 0.5 Pa by introducing pure Ar gas. The CP Ti substrate was etched by ion spurring with 300 V negative bias voltage and 2 A current for 15 min to remove the oxide layer of the substrate surface. Finally, A 400 V negative bias voltage and 3 A current was applied to the graphite target to prepare a single C layer, which was maintained for 2000 s. During the C film deposition process, the flow of Ar and duty cycle were maintained at 40 sccm and 20%, respectively. Then, the single C coating specimens were sealed in a quartz tube without oxygen, and were heated in a muffle furnace at 700 °C for 210 s. The schematic illustrations of the preparation process of the nanocomposite coating is shown in Figure 2. In this work, the coating is recorded as a single C coating, and coating + heating is recorded as C/TiC nanocomposite coating.

### 2.2. Characterization

The phase composition of the as-prepared all-coated Ti bipolar plates were characterized by X-ray diffraction (XRD), and the surface morphologies of coated specimens were observed using scanning electron microscopy (SEM). Raman spectroscopy was recorded with a 532 nm excitation wavelength to analyze the structural arrangements of the C film. In order to characterize the chemical and binding energy states, X-ray photoelectron spectroscopy (XPS) was applied to characterize the chemical and binding energy states.

### 2.3. Electrochemical Measurements

The electrochemical corrosion test was carried out by a CHI760e electrochemical workstation with a traditional three-electrode system. The BPs specimen was served as the working electrode, the platinum sheet was used as counter electrode and the saturated calomel electrode (SCE) acted as the reference electrode, respectively. The BPs specimen exposed to the electrolyte was a circle with an area of 1 cm$^2$, and the electrolyte solution was 0.5 M $H_2SO_4$ + 5 ppm HF solution and temperature was 80 °C to simulate the aggressive PEMFC environment. Potentiodynamic polarization, a potential scanning rate of 2 mV s$^{-1}$,

was adopted to evaluate the corrosion resistance of CP Ti and coated-Ti substrate. Before potentiodynamic polarization, the open circuit potential (OCP) was operated for 60 min. An Electrochemical Impedance Spectroscopy (EIS) test was carried out at OCP in a frequency from 100 kHz to 0.01 Hz. The EIS measurement results were further analyzed using curve fitting with ZView software.

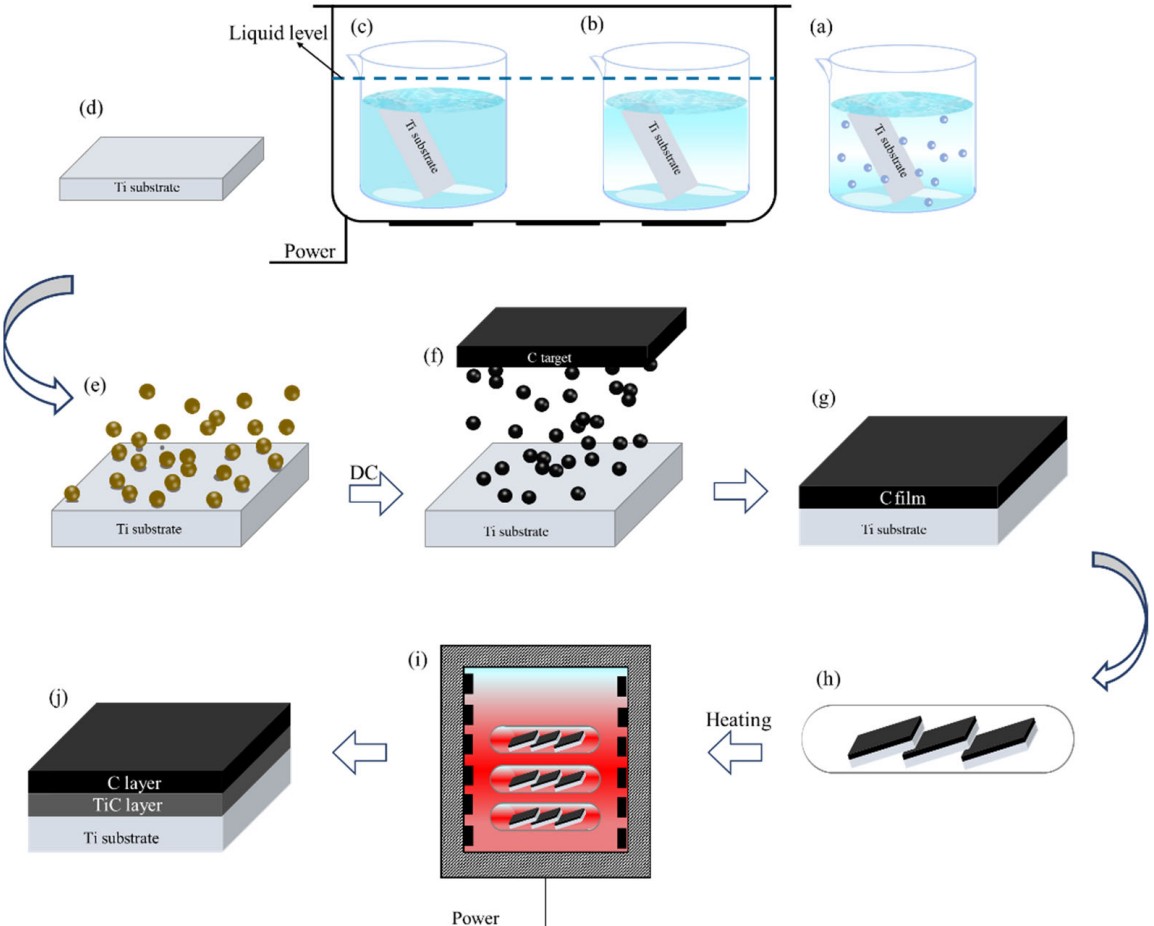

**Figure 2.** Schematic illustrations of preparation process of the nanocomposite coating: (**a**–**b**) preprocessing Ti substrate (including (**a**) Ti plate was immersed in 5 vol% HF solution; (**b**) and (**c**) Ti plate was cleaned ultrasonically with deionized water and alcohol, respectively; (**d**) dry Ti plate), (**e**–**g**) preparation process of single C layer (including (**e**) plasma cleaning; (**f**) depositing single C layer; (**g**) single C coating plate), (**h**–**j**) vacuum heat-treatment process (including (**h**) C coating Ti plates were sealed in quartz tube without oxygen; (**i**) the quartz tubes with single C coating plate were heated in muffle furnace; (**j**) C/TiC nanocomposite coating plate).

### 2.4. Interfacial Contact Resistance

In PEMFC stack, the ICR between the specimen and carbon paper (TGP-H-060, TORAY, Tokyo, Japan) was directly measured using a common method. The schematic diagram of the ICR measurement is shown in Figure 3; the test system is a sandwiched structure which was consists of two pieces of carbon paper, two Au-coated copper electrodes and a bipolar plate specimen [16,38,39]. The total resistance was measured at different pressures from 0.2 MPa to 2.0 MPa, and the ICR value of the typical bipolar plate specimen corresponded with its real value at a loading pressure of 1.4 MPa. The three specimens which had the same preparation conditions were tested to take the average value. The ICR can be calculated according to Equations (4)–(6) [8,16,40–42].

$$R_1 = 2(R_{Cu} + R_{Cu/C} + R_{C/BP} + R_C) + R_{BP} + R_0 \tag{4}$$

$$R_1 = 2(R_{Cu} + R_{Cu/C} + R_{C/BP} + R_C) + R_{BP} + R_0 \qquad (5)$$

$$ICR = \frac{S}{2}(R_1 - R_2) \qquad (6)$$

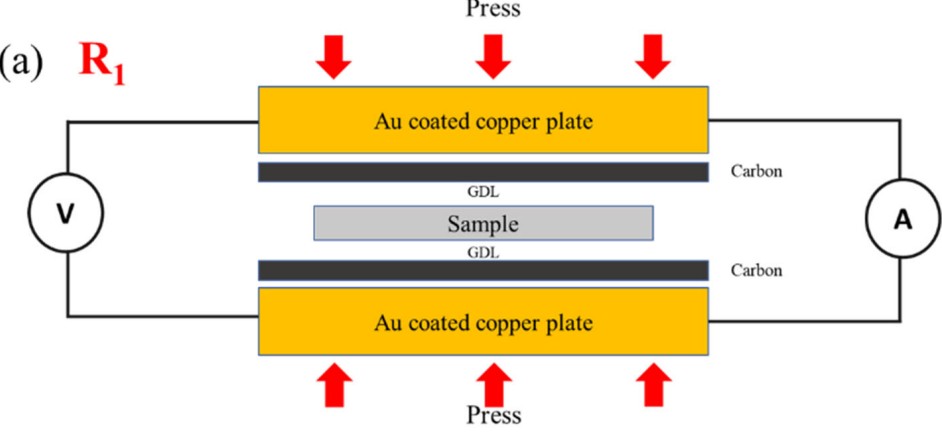

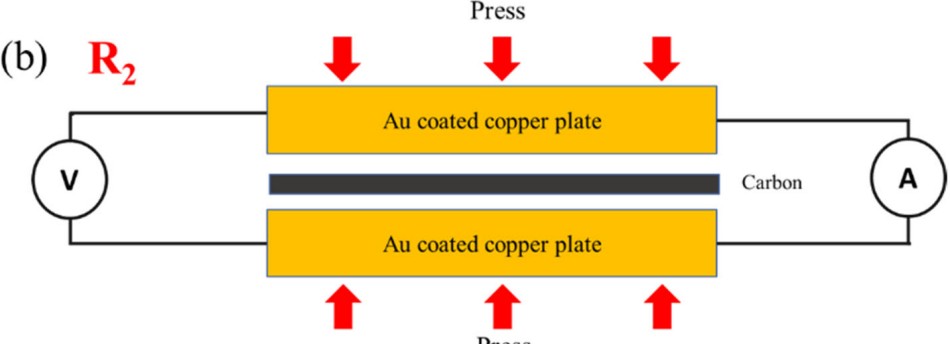

**Figure 3.** Schematic diagram of the ICR measurement. (**a**) the total resistance value of a bipolar plate; (**b**) the value of a carbon paper.

Where $R_1$ is the total resistance value (shown in Figure 3a), $R_{Cu}$ and $R_C$ is the bulk resistance of Au-coated copper electrode and carbon paper and $R_{Cu/C}$ is Contact resistance between them. $R_{BP}$ and $R_0$ represent the bulk resistance of bipolar plate and external system resistance, respectively. $R_{C/BP}$ (ICR) and S is Corresponding for interfacial contact resistance and contact surface area between specimen and carbon paper, respectively. While the $R_C$ and $R_{BP}$ were neglected because of their slight value.

## 3. Results and Discussion

### 3.1. Surface Structure and Morphology

Figure 4 shows the surface morphology of the as-prepared C coating and C/TiC nanocomposite coating on the titanium plate. It can be seen that the thickness of the C film is approximately 48.38 nm (illustration in Figure 4a), and the coating is relatively uniform judging from Figure 4a to Figure 4d. The surface morphology of specimens exhibit honeycomb-like structure, and there are some C nanoparticle clusters on the honeycomb edge (shown in orange dashed circle), which may result in reducing the ICR of bipolar plates because the graphite C possess great conductivity. As shown in Figure 4e, after vacuum heat treatment, it can be seen that some nanoparticles (red arrow) were embedded in the C film, which may be TiC particles.

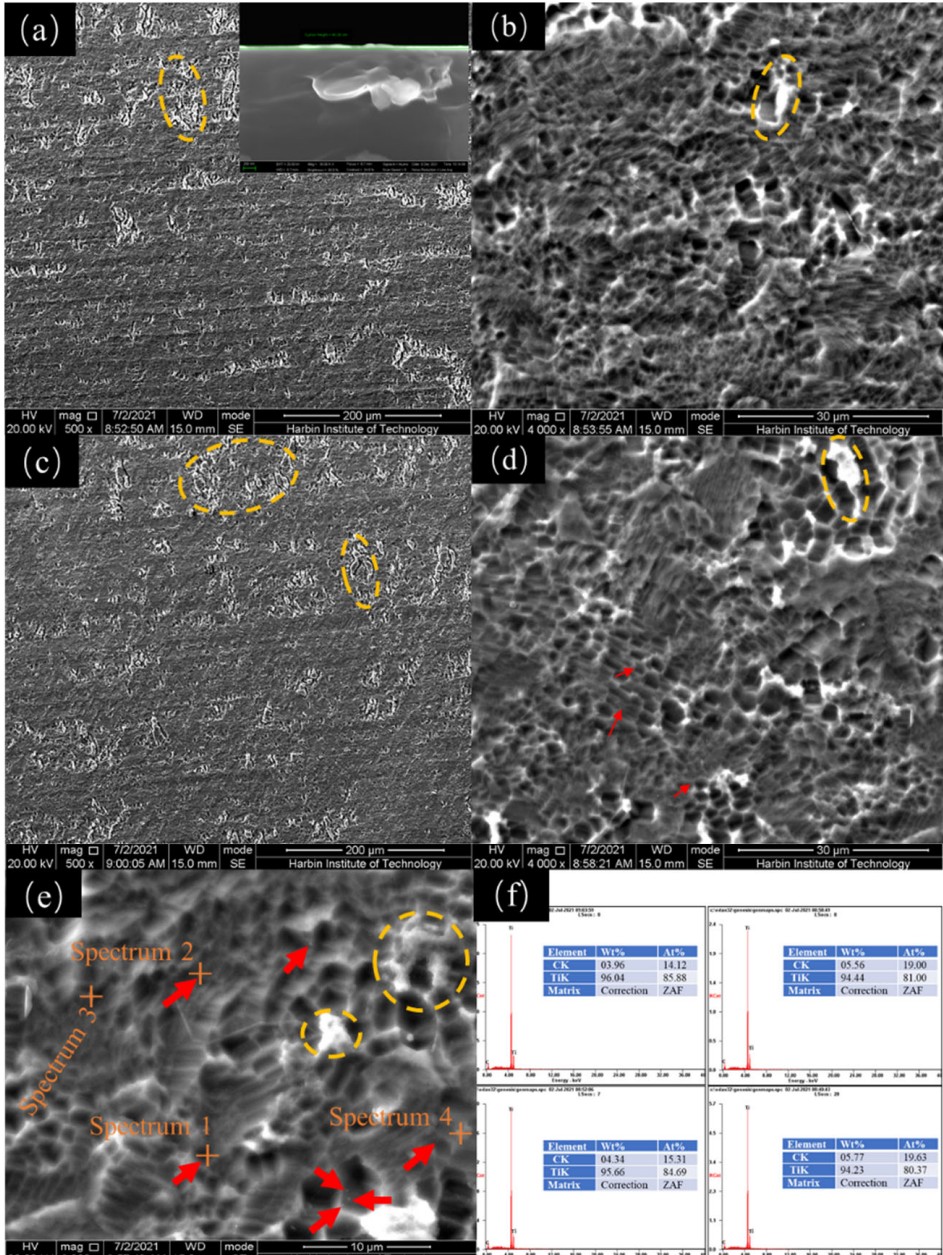

**Figure 4.** SEM images showing the surface morphology of (**a**,**b**) coating before vacuum heat treatment; (**c**,**d**) coating after vacuum heat treatment; (**e**) 8000× of coating after vacuum heat treatment; (**f**) EDX spectrum of coating after vacuum heat treatment.

　　　The composition of the nanoparticles at the orange cross-mark in Figure 4e was analyzed by EDX (Figure 4f). The peak of main ingredients are the minor C peak (~17.02 at.%) and the strong Ti peak without other element signal peaks. Thus, it can be inferred that TiC nanoparticles may exist in the as-prepared specimen, which is formed from the chemical reaction of active titanium atoms and carbon atoms during heat treatment. Additionally, the TiC interface coating thickness is roughly calculated in XPS.

　　　The crystal structure of the CP Ti and the as-prepared film's titanium bipolar plates were analyzed by X-ray diffraction pattern (XRD). The XRD spectra of CP titanium substrates, deposition single C coating and C/TiC nanocomposite coating titanium substrates are shown in Figure 5. The titanium substrate (CP Ti) has only peaks corresponding to hexagonal α-Ti. The peaks representing Ti, $TiO_2$ and C can be observed in the spectra of the single C coating. The TiC phase can be detected in the coating after heat treatment,

which means the TiC was formed during heat treatment. However, the peak of $TiO_2$ was not detected after heat treatment in Figure 5, because a carbothermal reduction of $TiO_2$ happened and titanium carbide was prepared according to the general equation [43,44]:

$$TiO_2 + 3C \rightarrow TiC + 2CO \tag{7}$$

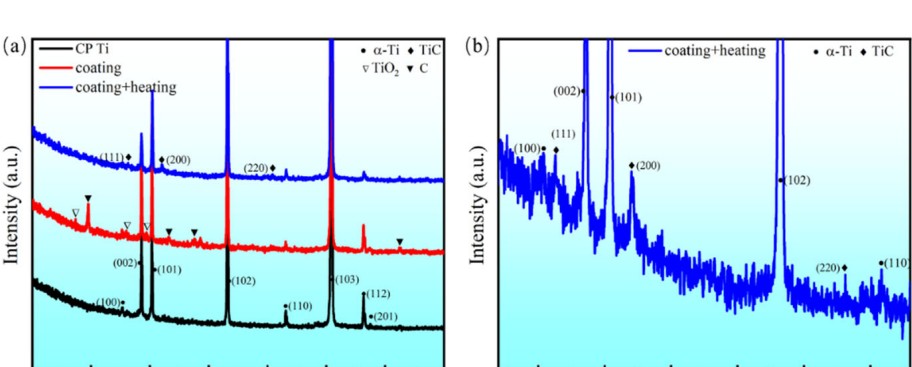

**Figure 5.** X-ray diffraction of (**a**) CP Ti, single C coating and C/TiC nanocomposite titanium substrate; (**b**) partial enlargement of C/TiC nanocomposite titanium substrate.

### 3.2. Composition of the Coating

In order to further analyze the film's composition and the carbon bonding states, the surface coatings before and after heat treatment were characterized by XPS. The XPS results were obtained by fixed 20% Lorentzian and 80% Gaussian [45] using XPS peak 4.1 software, as presented in Figure 6. It can be seen that the spectra were divided into four peaks, i.e., C-C ($sp^2$), C-C ($sp^3$), C-O and C=O [42,45,46]. The $sp^2$ bond fraction of the C/TiC nanocomposite coating was higher than $sp^3$; the area ratio of $sp^2$ to $sp^3$ in the C coating and C/TiC nanocomposite coating are 1.20 and 1.40, respectively. It can be explained that the $sp^2$ content increased due to the graphitization of $sp^3$ in carbon films with an increase in the temperature [47], the metastable carbon atom hybrid transitioned from $sp^3$ to $sp^2$, which means that the C/TiC nanocomposite Ti bipolar plate has better conductivity.

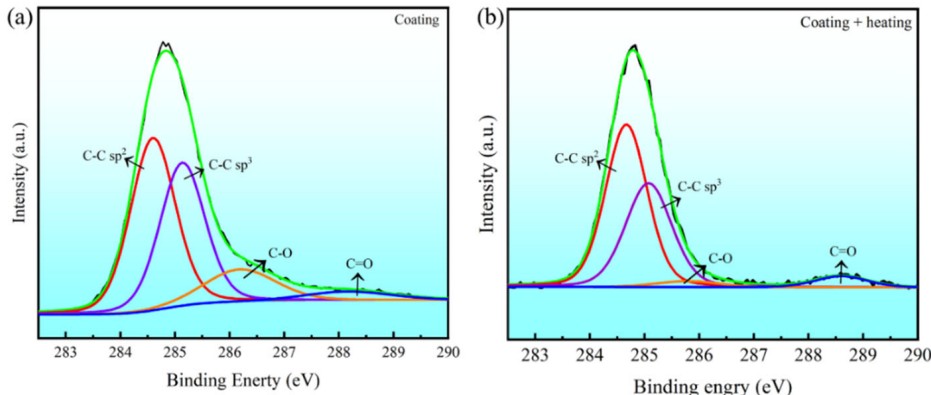

**Figure 6.** C 1 s photoelectron spectra of surficial C films: (**a**) C coating bipolar plate; (**b**) C/TiC nanocomposite bipolar plate.

In order to prove the TiC layer formed during vacuum heat treatment, Figure 7 presents the XPS C 1 s depth profile results for coatings before and after vacuum heat treatment. From Figure 7a, no obvious characteristic peaks of TiC could be seen and only C-C peaked before vacuum heat treatment. However, a weak peak of Ti-C bonding appeared at 90 s etching, which corresponded to the bonding energy of 281.8 eV [48], and the peak intensity increased with increasing etching time, as shown in Figures 7b and 8. This indicates that

TiC may be formed by activated Ti and C atoms, which absorb external energy during heat treatment [49,50]. We performed analysis for each sputtering level using detail by the XPS spectra of C 1 s, and the fitted depth profiles are shown in Figure 8a–j. It was observed that the Ti-C did not appear at the first two 90 s etching, and appeared at 90 s with a negligible level (0.79%). By increasing the etching time, the Ti-C bonding fractions increased, i.e., 8.03%, 6.17%, 7.03%, 8.08% and 8.87% corresponding to 120 s, 150 s, 180, 210 s, 240 s and 270 s, respectively. And the thickness of the TiC interface layer is approximately 19.5 nm by rough estimation because of its 0.32 nm/s etching rate. A certain thickness of the intermediate layer can improve the adhesion strength between the substrate and the surface C layer [21,51–54], forming the schematic diagram shown in Figure 9. Moreover, C-O, C-C bonding fractions decreased to varying degrees with increased etching time and show dynamic distribution. Such a thin intermediate layer and bigger $sp^2$ bond fraction would improve the adhesion strength and surface conductivity, providing a high binding force and low ICR.

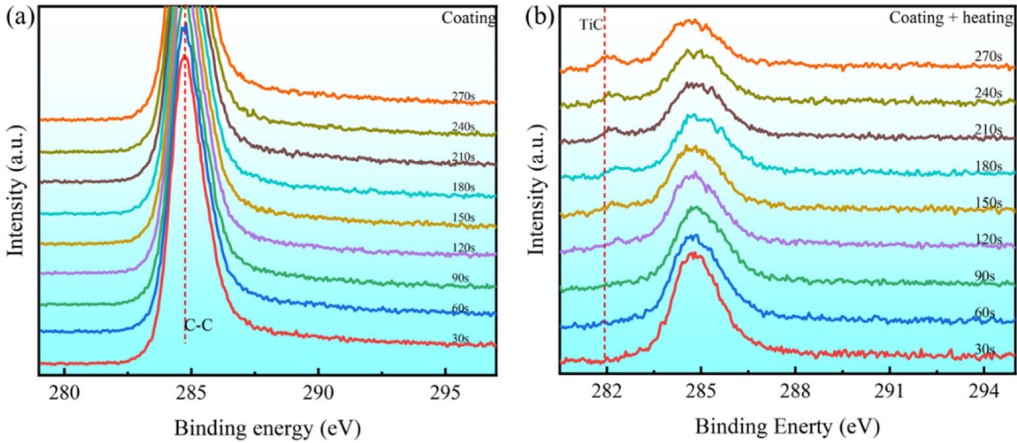

**Figure 7.** XPS spectra of C 1 s the depth profile (**a**) C coating; (**b**) C/TiC nanocomposite coating.

### 3.3. Raman Spectroscopy

Figure 10 exhibited the Raman spectra of the film before and after vacuum heat treatment. Two asymmetric characteristic peaks, D and G peaks, are seen at wavenumbers in the range 1100 cm$^{-1}$ to 1800 cm$^{-1}$, which are centered at ~1380 cm$^{-1}$ and ~1560 cm$^{-1}$, respectively. The D-band is caused by $sp^2$ ring of defects and impurities from disordered structure in the graphite lattice, and the G-band represents plane stretching vibration $sp^2$ hybrid carbon, both rings and chains [55]. As shown in Figure 10c,d, the C film structure was analyzed by Gauss fitting in detail, with the results listed in Figure 10b. Compared to single the C coating, the $I_D/I_G$ value of C/TiC nanocomposite coating (4.43) is higher than the C coating (3.78) and the G peak position moves towards higher wavenumbers; this indicates that the C/TiC nanocomposite coating specimen has a higher ratio of $sp^2/sp^3$ and a higher $sp^2$ content, and means that the higher $sp^2$ content of the coating has lower ICR [22,40]. This result is consistent with Zhang et al. [16] and XPS (Figure 6). In addition, FWHM before and after heat treatment were 109.04 and 103.39 in this work, respectively. After vacuum heat treatment, C atom lattice stress was formed in the C film, leading to an instable film structure. At this time, the C atom will undergo migration and rearrangement to reduce the degree of distortion in the bond angle and bond length of $sp^2$ cluster. This is beneficial to release internal stress and to change the instable film structure [56,57], and the lowering of internal stress can promote $sp^2$ phase formation and improve conductivity of bipolar plates [58]. That is to say, the lower FWHM, the higher conductivity, corresponding with XPS results.

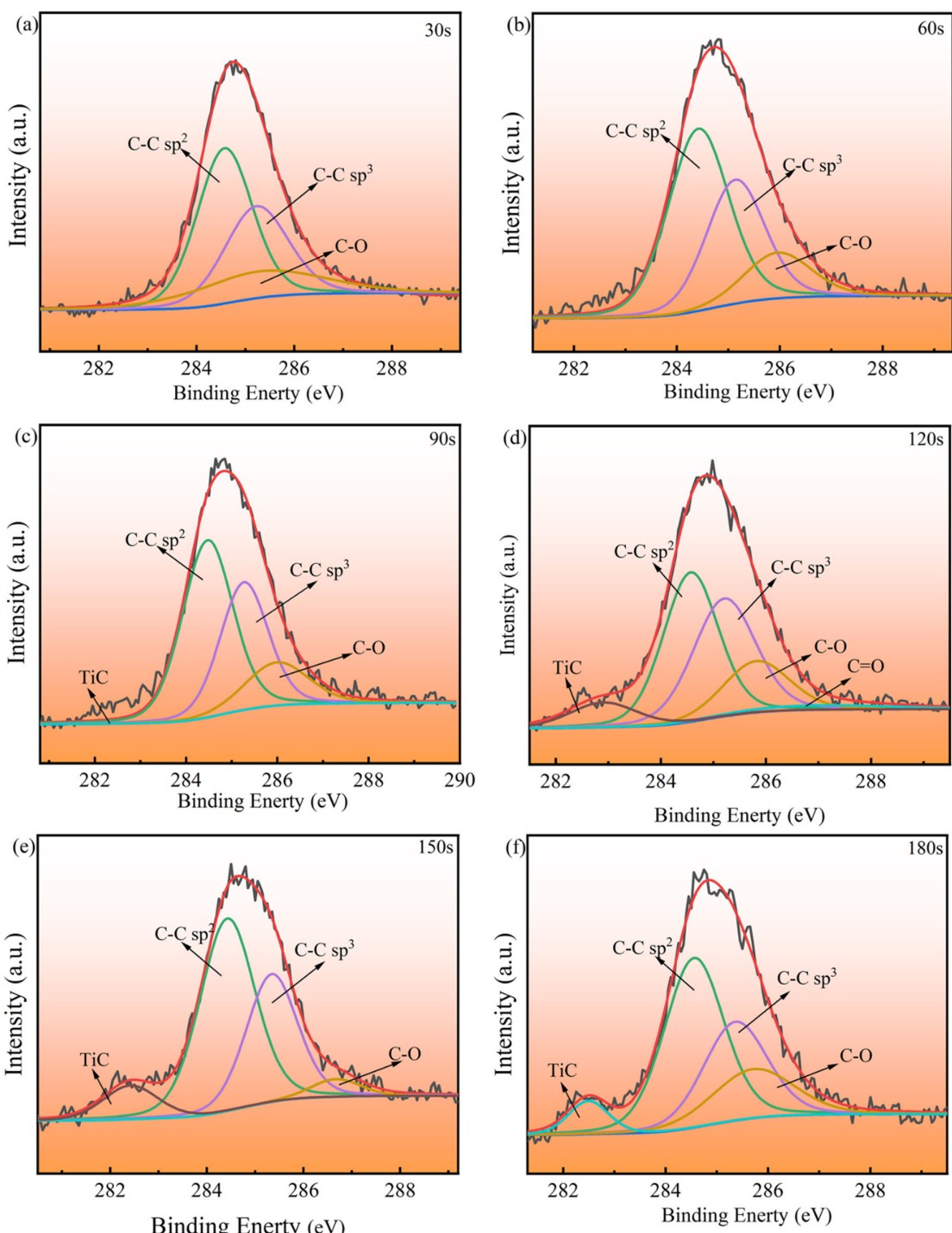

**Figure 8.** *Cont.*

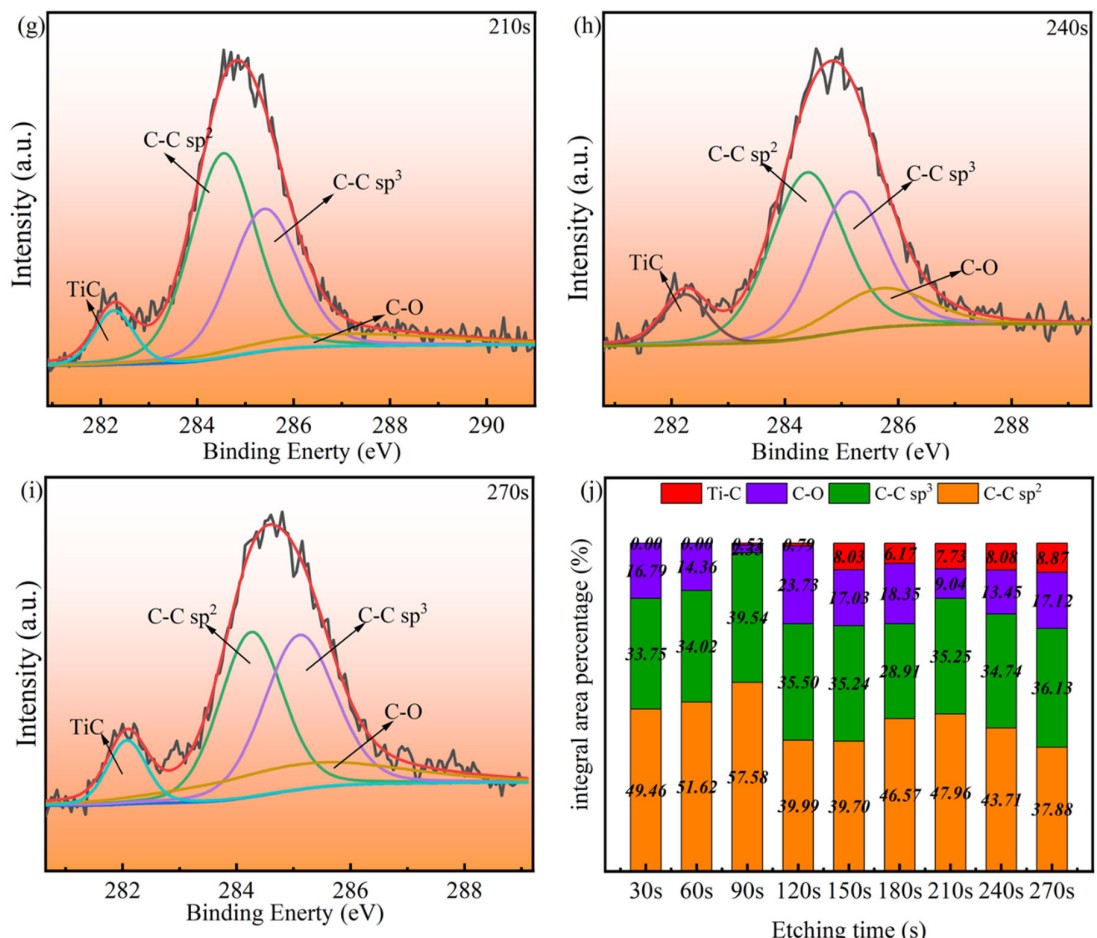

**Figure 8.** XPS spectra of C 1 s depth profile of C/TiC nanocomposite coating; (**a–i**) the depth profile of C/TiC nanocomposite coating at 30 s, 60 s, 90 s,120 s,150 s,180 s, 210 s, 240 s, 270 s, respectively; (**j**) the results of bond fractions measured.

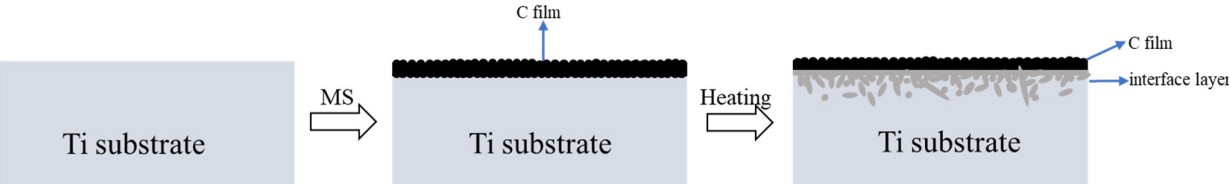

**Figure 9.** Schematic of the film formation process.

### 3.4. Electrochemical Property

Figure 11a shows the variation of open circuit potential (OCP) of the CP Ti and as-prepared specimens with immersion time in simulated PEMFCs. It can be seen that the OCP of the CP Ti and C coating specimens decreases rapidly during the initial immersion period and then stabilizes at relatively lower potentials. However, compared with the former, the C/TiC nanocomposite coating specimen remains relatively stable throughout the immersion period with a higher potential. The OCP results of the three specimens are listed in Table 1. The potential of three specimens are −0.16 V, 0.04 V and 0.18 V, respectively, following the order of C/TiC nanocomposite coating > C coating > CP Ti. Thermodynamically, the higher potential represents higher chemical stability [59], which indicates that the C/TiC nanocomposite coating specimens have better corrosion resistance in simulated PEMFCs.

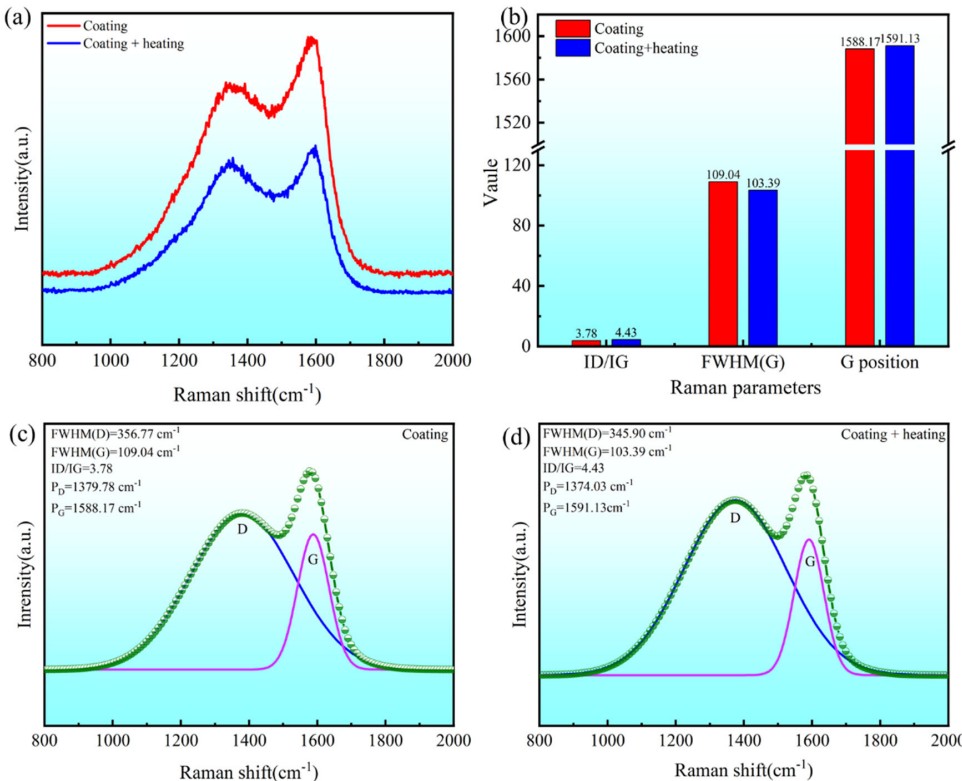

**Figure 10.** Raman of the C coating and C/TiC nanocomposite coating. (**a**) the Raman spectroscopy of C coating and C/TiC nanocomposite coating; (**b**) the results of ID/IG, FWHM(G) and G position; (**c**,**d**) the curve fitting of C coating and C/TiC nanocomposite coating.

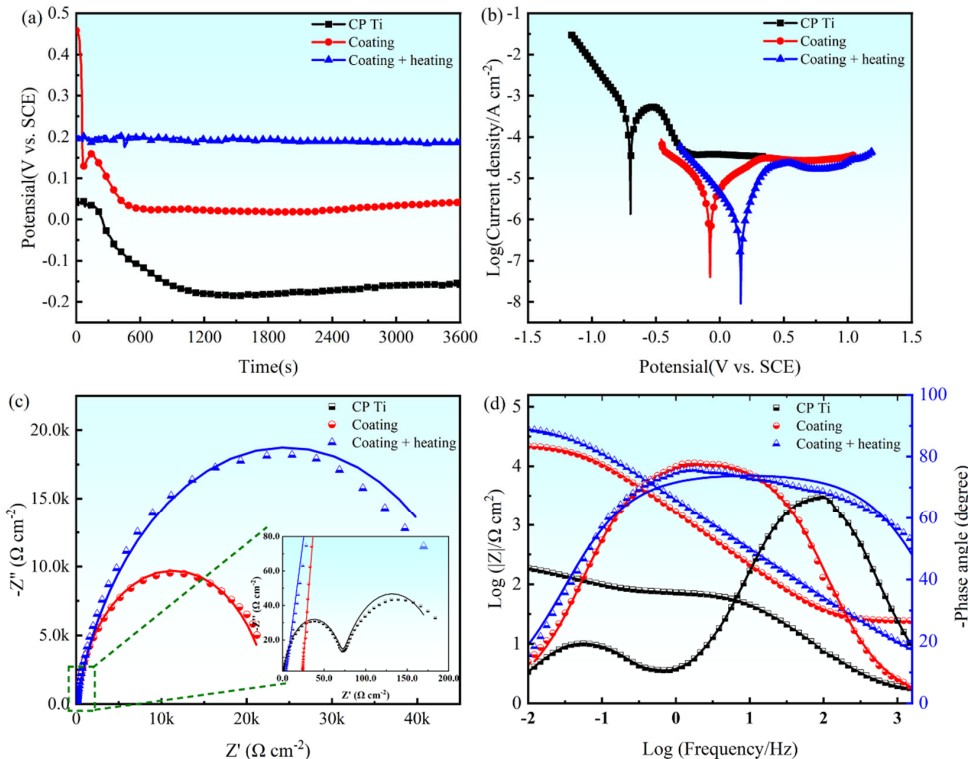

**Figure 11.** The electrochemical test curves of CP Ti, C coating and C/TiC nanocomposite coating: (**a**) OCP; (**b**) potentiodynamic polarization curve; (**c**) Nyquist plot of EIS; (**d**) bode plot of EIS.

**Table 1.** Corrosion parameters and EIS fitted results for CP Ti and coated Ti specimen.

| Specimen | $R_s$ ($\Omega$ cm$^2$) | $Q_f$ ($\Omega^{-1}$ s$^{n1}$ cm$^2$) | $R_f$ ($\Omega$ cm$^2$) | $Q_{dl}$ ($\Omega^{-1}$ s$^{n1}$ cm$^2$) | $R_{ct}$ ($\Omega$ cm$^2$) | OCP (V) | $E_{corr}$ (V) | $i_{corr}$ ($\mu$A cm$^{-2}$) |
|---|---|---|---|---|---|---|---|---|
| CP Ti | 1.63 | - | 71.14 | - | 69.51 | −0.16 | −0.70 | 345.10 |
| C Coating | 23.59 | $1.14 \times 10^{-4}$ | 22,736.57 | $1.14 \times 10^{-4}$ | 22,713 | 0.04 | −0.08 | 3.18 |
| C/TiC Coating | 3.56 | $7.96 \times 10^{-5}$ | 49,036.56 | $7.96 \times 10^{-5}$ | 49,033 | 0.18 | 0.16 | 0.74 |

The corrosion current density is one of the most important parameters for evaluating the performance of BPs in a fuel cell environment. Figure 11b shows the potentiodynamic polarization curves of CP Ti and as-prepared specimens (including the C/TiC nanocomposite coating and C coating) Ti bipolar plates. It is clear that the CP Ti substrates undergo an activation–passivation transition region and then enter the passivation region as the potential increases. The transition region may consist of an oxidation process which always occurs at the beginning of passivation and prevents the corrosion from continuing [30]. Furthermore, the polarization curves of the C coating and C/TiC nanocomposite coating specimens show typical Tafel behavior without a transition region, because a dense C film was prepared on the Ti plate surface before corrosion test. As shown in Table 1, the corrosion current density and corrosion potential of the three specimens are 345.10 $\mu$A cm$^{-2}$, 3.18 $\mu$A cm$^{-2}$, 0.74 $\mu$A cm$^{-2}$, −0.7 V, −0.08 V, and 0.16 V, respectively. In the results, the corrosion potential follow the order of C/TiC nanocomposite coating > C coating > CP Ti, and corrosion current density follows the opposite of this sequence. Tables 2 and 3 show the values of $E_{corr}$ and $i_{corr}$ for various coatings. Generally, lower corrosion current density and higher corrosion potential means better chemical inertness and higher corrosion resistance [60]. Moreover, when comparing with CP Ti, corrosion current density of the C/TiC nanocomposite coating specimen enormously decreased by about three orders of magnitude, which met the target requirement of 2020 DOE (1 $\mu$A cm$^{-2}$). Comparing the $i_{corr}$ values between the specimens before and after heat treatment, the C coating is larger than the C/TiC nanocomposite coating specimen; this may be due to the formation of the TiC phase during heating, which has better corrosion resistance. Table 3 shows the $i_{corr}$ values of various coatings. It can be found that the $i_{corr}$ value of as-obtained C/TiC nanocomposite coating is higher than for noble metal Ag doped coating [16], metal Nitride- [27] and metal carbide- [23] coating, but is much less than Zr [31], $\alpha$-C [23], F-doped zinc tin oxide [4], TiN [19] and Cr-C/a-C:Cr coating [53], which indicates that the C/TiC nanocomposite coating in this work shows high corrosion resistance in PEMFCs environment.

**Table 2.** The Ecorr values of various coating for bipolar plate.

| Coating | Ecorr/V | Ref. |
|---|---|---|
| a-C/316L | 0.2~0.3 | [54] |
| a-C/TA2 | 0.254 | [37] |
| a-C/Ti/316L | 0.175~0.355 | [40] |
| (Ti,Zr)N–Ti | 0.17 | [27] |
| C/TiC nanocomposite coating | 0.16 | This work |
| a-C/316L | 0.138 | [37] |
| ZrCN/TC4 (2 ppm) | 0.09 | [34] |
| ZrCN/TC4 (4 ppm) | 0.06 | [34] |
| ZrCN/TC4 (6 ppm) | 0.04 | [34] |
| TiN-316L | 0.02 | [19] |
| CrN-316L | −0.018 | [19] |
| TiO$_2$/Ti/Al1050 | −0.054 | [3] |

**Table 3.** The $i_{corr}$ values of various coating for bipolar plate.

| Coating | Electrolyte | $i_{corr}$ ($\mu A\ cm^{-2}$) | Ref. |
|---|---|---|---|
| Zr/TC4 | 0.5 M $H_2SO_4$ + 2 ppm HF, 70 °C | 7.46 | [31] |
| SnOx:F | 1 M $H_2SO_4$ + 2 ppm HF, 70 °C | 6.64 | [4] |
| $\alpha$-C | 0.5 M $H_2SO_4$ + 5 ppm HF, 70 °C | 3.56 | [23] |
| TiN/316L | 0.1 M $H_2SO_4$ + 2 ppm HF, 80 °C | 2.5 | [19] |
| ZnSnOx:F | 1 M $H_2SO_4$ + 2 ppm HF, 70 °C | 1.2 | [4] |
| ZrCN | 0.5 M $H_2SO_4$ + 6 ppm HF, 70 °C | 0.985 | [34] |
| Cr-C/a-C:Cr | 0.5 M $H_2SO_4$ + 5 ppm HF, 70 °C | 0.785 | [53] |
| C/TiC nanocomposite coating | 0.5 M $H_2SO_4$ + 5 ppm HF, 80 °C | 0.74 | This work |
| Zr-C/$\alpha$-C | 0.5 M $H_2SO_4$ + 5 ppm HF, 70 °C | 0.49 | [23] |
| (Ti,Zr)N–Ti | 0.5 M $H_2SO_4$ + 3 ppm HF, 60 °C | 0.212 | [27] |
| Ag:Cr/$\alpha$-C | pH = 3, 0.1 ppm HF, 80 °C | 0.15~0.38 | [16] |

Electrochemical Impedance Spectroscopy (EIS) was performed to describe the corrosion process of specimens in 0.5 M $H_2SO_4$ + 5 ppm HF solution at 80 °C. As shown in Figure 11c–d, EIS was tested under 1 h OCP in a simulated PEMFC environment, which provided a steady state of potential. The Nyquist plot (Figure 11c) of the C coating and C/TiC nanocomposite coating specimens shows a single capacitive arc across the entire frequency range, the C/TiC nanocomposite coating specimens possess a larger impedance gradient and radius, while the two capacitive loops can be observed in the CP Ti. Furthermore, in the bode plot of the three specimens (Figure 11d), the phase angles of the specimens before and after heating approaches 80°, larger than the CP Ti (~65°) from medium to low frequencies. This indicates that a highly stable and compact film forms on the surface which can improve the corrosion resistance under PEMFCs conditions [61]. Generally, the larger impedance gradient, radius and phase angle, the better the corrosion resistance [40]. It can be concluded that the order of corrosion resistance is C/TiC nanocomposite coating > C coating > CP Ti, which is consistent with results of the potentiodynamic polarization test.

The Electrochemical Impedance Spectroscopy was analyzed by equivalent circuits using the ZView software, as shown in Figure 12. According to the characteristics of the three specimens in this work, the equivalent circuit with one time constant (Figure 12a) was applied to the coating and the C/TiC nanocomposite coating Ti plate, and two time constant (Figure 12b) was applied to CP Ti to analyze their corrosive behavior. Among the equivalent circuits, $R_s$, $R_f$ and $R_{ct}$ corresponded to the resistance of the solution, the pore resistance in the film and the charge transfer resistance, respectively [20]. In addition, a constant phase element (CPE) was introduced to describe non-ideal capacitance of passivation film or C film. The $Q_{dl}$ and $Q_f$ represent electric double layer capacitance and the capacitance of passivation film or C film [59]. The constant phase angle element impedance can be calculated by Equation (8):

$$Z_Q = [Y_0(j\omega)\,n]^{-1} \tag{8}$$

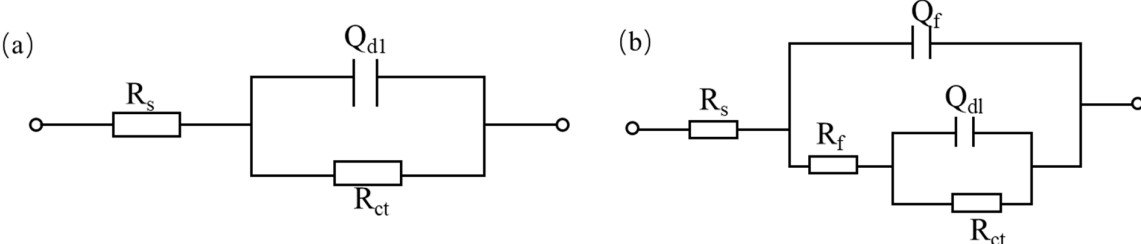

**Figure 12.** Equivalent electrical circuits used for modeling experimental EIS data of the (**a**) C coating and C/TiC nanocomposite coating and (**b**) CP Ti.

$Z_Q$ and $Y_0$ are the impedance and admittance of the constant phase angle element Q, respectively. $\omega$ is the angular frequency of the excitation signal, $n$ is the dispersion index. Table 1 shows the fitted results for the three specimens. It is well known that the polarization resistance is inversely proportional to the corrosion rate and is regarded as the sum of $R_{ct}$ and $R_f$. The $R_p$ value of the C/TiC nanocomposite coating is 49,036.56 $\Omega$ cm$^2$, which is larger than the C coating (22,736.59 $\Omega$ cm$^2$) and CP Ti (71.14 $\Omega$ cm$^2$) descending in order, C/TiC nanocomposite coating, C coating and CP Ti. This indicates that the corrosion resistance of the coating improved after vacuum heat treatment. The results are consistent with the potentiodynamic polarization analysis.

*3.5. Conductivity*

Interfacial contact resistance (ICR) is a key parameter to measure the conductivity of a material that directly affects the output power of the bipolar plate. In general, the ICR between GDL and bipolar plates is inversely proportional to the power density and performance of PEMFC. The ICR, CP Ti, C coating and C/TiC nanocomposite coating specimens were investigated using the method reported by Wang et al. [39] (Figure 3), and it was calculated by Equations (4)–(6); the results of ICR was shown in Figure 13. By increasing the pressure, the ICR value of all tested specimens decreased. The ICR dropped significantly in the low pressure area because the effective contact area between GDL and BPs increased with the compaction force. However, ICR keeps relatively steady at high pressure range; this relates to roughness and chemical components of bipolar plates. It can be observed that the C/TiC nanocomposite coating bipolar plate exhibits lower ICR than a single C coating and CP Ti bipolar plate. Under pressure of 1.4 MPa, the ICR of C/TiC nanocomposite coating and C coating is 2.34 m$\Omega$ cm$^2$ and 5.34 m$\Omega$ cm$^2$, respectively. Compared to the single C coating Ti, ICR of C/TiC nanocomposite coating Ti decreased 2 times, which meets 2020 DOE target level (10 m$\Omega$ cm$^2$). The result was consistent with Raman and XPS. However, a single C coating had a highest ICR value among the three specimens, and it may have the roughest surface [25,62] and the lowest ID/IG, which could be seen from SEM, Raman and XPS. It must be pointed out that the ICR of the CP Ti (black solid line) is lower, which was pre-treated to remove the surface oxide film (TiO$_2$). The ICR of raw pure Ti (purple solid line in illustration) is 38.66 m$\Omega$ cm$^2$, as shown in the inset of Figure 13. It well known that TiO$_2$ possesses semiconductor properties to increase the ICR of the bipolar plate, and directly affects the conductivity of the battery. Moreover, compared to the ICR values in Table 4, it shows that the ICR value of the as-obtained C/TiC nanocomposite coating is much lower than that for noble-doped metal coating [29], a-C coating [13,20,21,37,40–42,52], metal carbide and nitride coating [8,19,23,34,35,38,53,59,60,63], but is less than Ag-doped coating [30], a-C coating [52,54], metal carbide and nitride coating [36]. So, this work is significant to improve the conductivity of bipolar plates in PEMFCs.

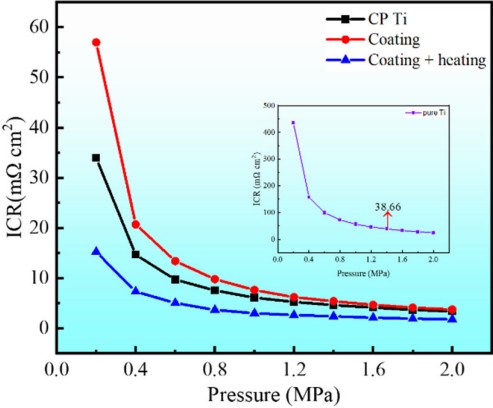

**Figure 13.** ICR of CP Ti, C coating and C/TiC nanocomposite coating.

**Table 4.** The ICR values of various coatings for bipolar plates.

| Coating | ICR | Ref. |
|---|---|---|
| a-C/316L | 36.1 | [42] |
| CrN/316L | 23 | [19] |
| TiC/Ti | 20.9 | [35] |
| a-C:H/316L | 17.6 | [21] |
| TiCrC/Ti | 16.5 | [35] |
| TDMAT-TiN/316L | 15.239 | [63] |
| Carbon-coated 316L | 12 | [41] |
| ZrCN/Ti | 11.2 | [34] |
| TiN/316L | 10 | [19] |
| Nb-N/304 | 9.26 | [38] |
| Nb-C/304 | 8.47 | [60] |
| TiC-TA1 | 7.5 | [8] |
| $\alpha$-C/TA2 | 6.52 | [37] |
| $\alpha$-C/316L | 5.64 | [37] |
| a-C/304 | 5.4 | [13] |
| CrTiN-4A/316L | 4.57 | [59] |
| a-C/TiC$_x$/316L (60V) | 4.55 | [52] |
| Ti-Ag/Ti | 4.1 | [29] |
| a-C/TiC$_x$/316L | 3.85~4.27 | [40] |
| Zr-C/a-C/316L | 3.63 | [23] |
| Cr-C/a-C:Cr/316L | 2.89 | [53] |
| $\alpha$-C/316L (H7) | 2.53 | [40] |
| C/TiC nanocomposite coating | 2.34 | This work |
| Ti-Ag-N/Ti | 2.3 | [30] |
| a-C/TiC$_x$/316L (60V/300V) | 1.85 | [52] |
| a-C/TiC$_x$/316L (300V) | 1.92 | [52] |
| a-C/TiC$_x$/316L | 1.93 | [54] |
| Nb-Cr-C/TA1 | 1.15 | [36] |

## 4. Conclusions

In this study, the C/TiC nanocomposite coating has been prepared on the surface of titanium metallic by magnetron sputtering technology and vacuum heat treatment technology. This prepared C/TiC nanocomposite coating was characterized by SEM, XRD, XPS, Raman, electrochemical testing and ICR. The result is as follows:

1. A C/TiC nanocomposite coating has been successfully prepared on the surface of titanium metallic substrate by combination of magnetron sputtering technology and vacuum heat treatment technology, which consists of a C surface layer (~28.88 nm) and TiC interface layer (~19.5 nm).

2. The corrosion resistance of the C/TiC nanocomposite coating titanium bipolar plate was investigated in 0.5 M $H_2SO_4$ + 5 ppm HF solution at 80 °C. The results show that the corrosion resistance of the C/TiC nanocomposite coating (0.74 μA cm$^{-2}$) was greatly improved compared with that of commercially pure titanium substrate (345.10 μA cm$^{-2}$), and the corrosion current density of the C/TiC nanocomposite coating decreased by 3 orders of magnitude in a simulated cathodic environment.

3. Interfacial contact resistance of the C/TiC nanocomposite coating titanium bipolar plate is 2.34 mΩ cm$^2$ under 1.4 MPa compaction force, which is much lower than that of raw CP Ti (38.66 mΩ cm$^2$).

**Author Contributions:** Methodology, X.W., G.L., Y.F., D.S. and F.K.; investigation, G.L., Y.F. and F.K.; resources, D.S.; study design, W.M.; data analysis, W.M.; date collection, W.M.; writing—original draft preparation, W.M.; writing—review and editing, H.Z. and F.K.; supervision, H.Z., X.W. and F.K.; revision, H.Z.; Conceptualization, F.K. All authors have read and agreed to the published version of the manuscript.



**Funding:** This work was supported by National Key Research and Development Program (2020YFB1505903). This work was also supported by open project of State Key Laboratory of Vanadium and Titanium Resources Comprehensive Utilization (2020P4FZG08A). Their support is gratefully appreciated by the authors.

**Institutional Review Board Statement:** Not applicable.

**Informed Consent Statement:** Not applicable.

**Data Availability Statement:** Not applicable.

**Conflicts of Interest:** The authors declare no conflict of interest.

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
