# Peer review of "Electrochemical Behavior and Surface Conductivity of C/TiC Nanocomposite Coating on Titanium for PEMFC Bipolar Plate"

_metals, doi:10.3390/met12050771_

Round 1

Reviewer 1 Report

The paper is well-presented, carefully prepared, good length and the results convincingly support the conclusions. The manuscript gives interesting results and  I would recommend this manuscript for publication in Metals.

My general suggestion is to accept the paper after a minor improvement.

Comments:

Figures 7 and 10 should be improved (they are not clear, poor  visibility)

Language should be improved.

Author Response

Dear reviewer 

Thank you for your comments and suggestions on our manuscript. According to your comments, we have modified the manuscript and detailed corrections are listed in attached document.

Reviewer 2 Report

The topic is interesting and the manuscript can be published after a minor revision.   There are a number of points that authors should take into account.

i. The authors write in the Abstract: "The corrosion current density (icorr) decrease 4 orders of magnitude in simulated cathodic environment. " However, this contradicts the text in Section 3.4 (Comparing with CP Ti, corrosion current density of C/TiC nanocomposite coating specimen was enormously decreased about 3 orders of magnitude, which meets target requirement) and the data in Figure 10b.

ii.Page 3, 10-11 lines from above: "...and all the surface the were etched in the solution containing 5 vol% HF for 30 s in order... " Refine the text.

            iii. Page 7, 4 line from above: "However, after 90 s etching, the peak of Ti-C bonding was appeared bonding energy of approximately 281.8 eV" Refine the text.

iv. Page 7, 6-7 lines from above: "It indicating that TiC may be formed rapidly by absorbing external energy inducing Ti and C atomic activation during heat treatment [47, 48]." Refine the text.

v. Page 11, 7-8 lines from above: "As shown in Table 1, the corrosion current density and corrosion potential of three specimen are 345.10 μA cm-2, 3.18 μA cm-2, 0.74 μA cm-2, -0.7V, - 0.08V, and 0.16V, respectively. Which follow the order of C/TiC nanocomposite coating> C coating>CP Ti, generally, lower corrosion current density and higher corrosion potential means better chemical inertness and higher corrosion resistance " Authors should clarify whether the sequence "C/TiC nanocomposite coating> C coating>CP Ti.." refers to corrosion current density or corrosion potential.

vi. Table 1. The value of OCP for CP Ti does not correspond to Figure 10a.

            vii. Page 11, 5 line bottom: "the Rp of is C/TiC nanocomposite coating was 49036.56 Ω cm2, which is larger... Refine the text.

            viii. Figure 10. Instead of "The electrochemical test curves of CP Ti, C coating and C/TiC nanocomposite coating: (a) OCP; (b) potentiodynamic polarization curve; (c) bode plot of EIS; (d) Nyquist plot of EIS." should be  "(c) Nyquist plot of EIS, (d) bode plot of EIS"

"          ix. Page 13, 10 line from above. The authors write : "The ICR of raw pure Ti (purple solid line) is 38.66 mΩ cm2, as shown in Figure 12. " However, in Fig. 12 highest ICR value is shown for coating.

x. Conclusion 2. The authors write : "and corrosion current density of C/TiC nanocomposite coating decrease 4 orders of magnitude in simulated cathodic environment. However, this contradicts the text in Section 3.4 (Comparing with CP Ti, corrosion current density of C/TiC nanocomposite coating specimen was enormously decreased about 3 orders of magnitude, which meets target requirement) and the data in Figure 10b.

Author Response

Dear Reviewer

Sincerest thanks for your response and reviewers comments on our manuscript. According to your comments, we have modified the manuscript and detailed corrections are listed in attached document.

Reviewer 3 Report

In the article combination of magnetron sputtering technology and vacuum heat treatment technology was used to prepare C/TiC nanocomposite coated titanium bipolar plate. The article complements existing research and presents a way to realize a C/TiC nanocoated titanium bipolar plate with reduced costs and increased performance.

The article is interesting and timely. There do not seem to be any major shortcomings. Here are a few comments that I believe will further develop the article.

Abstract:

  • Very concise but includes what is essential in the article.

Introduction:

  • Summarise recent research on the subject and outline the need for research.

Experiments details / Materials and Methods:  

  • Commercial pure titanium (CP Ti) is a group containing four grades (1...4). Which CP Ti grade was used in the study?
  • The section "2.1. Preparation of the nanocomposite coating" could be further illustrated with a diagram.

Results and discussion:

  • The results are clearly presented.
  • The discussion is now in the context of the presentation of the results and is quite limited. The authors could consider expanding the discussion sections. The Discussion section could even be taken as a separate chapter.
  • Title “ results and discussion” – Capital R to word “results” ?

Conclusions: OK

Author Response

Dear Reviewers

Thank you for your comments and suggestions on our manuscript. According to your comments, we have modified the manuscript and detailed the corrections in attached document.

Reviewer 4 Report

This manuscript presents a new technology to prepare TiC/C coating to provide protectiveness of the PEMFC bipolar plate, which is interesting and helpful for the following related research. It can be accepted after revision.

(1) in the experimental section, the scan rate is 2mV/s in potentiodynamic curve which is too fast. The normal rate is 0.5 mV/s

(2) In Fig.3, it is difficult to measure the thickness of TiC/C coating. Please provide cross-sectional morphology. Moreover, it is better to provide EDX mapping result to show the structure of TiC/C coating.

(3) In Fig.4, why does the TiO2 disappear after heat treatment.

Author Response

Dear reviewer  

Thank you for reading our manuscript and reviewing it, which will help us improve it to a better scientific level. We revised our manuscript and changes have taken place.  For details, it can be seen in the attached document.

Round 2

Reviewer 4 Report

The mentioned concerns have been addressed by the authors. It can be accepted now.